# Shelf-Life Management and Ripening Assessment of ‘Hass’ Avocado (*Persea americana*) Using Deep Learning Approaches

**DOI:** 10.3390/foods13081150

**Published:** 2024-04-10

**Authors:** Pedro Xavier, Pedro Miguel Rodrigues, Cristina L. M. Silva

**Affiliations:** CBQF—Centro de Biotecnologia e Química Fina—Laboratório Associado, Escola Superior de Biotecnologia, Universidade Católica Portuguesa, Rua Diogo Botelho 1327, 4169-005 Porto, Portugal; pxavier@ucp.pt (P.X.); pmrodrigues@ucp.pt (P.M.R.)

**Keywords:** convolutional neural network, fruit ripening, shelf-life tracking, post-harvest handling, supply chain management

## Abstract

Avocado production is mostly confined to tropical and subtropical regions, leading to lengthy distribution channels that, coupled with their unpredictable post-harvest behavior, render avocados susceptible to significant loss and waste. To enhance the monitoring of ‘Hass’ avocado ripening, a data-driven tool was developed using a deep learning approach. This study involved monitoring 478 avocados stored in three distinct storage environments, using a 5-stage Ripening Index to classify each fruit’s ripening phase based on their shared characteristics. These categories were paired with daily photographic records of the avocados, resulting in a database of labeled images. Two convolutional neural network models, AlexNet and ResNet-18, were trained using transfer learning techniques to identify distinct ripening indicators, enabling the prediction of ripening stages and shelf-life estimations for new unseen data. The approach achieved a final prediction accuracy of 88.8% for the ripening assessment, with 96.7% of predictions deviating by no more than half a stage from their actual classifications when considering the best side of the samples. The average shelf-life estimates based on the attributed classifications were within 0.92 days of the actual shelf-life, whereas the predictions made by the models had an average deviation of 0.96 days from the actual shelf-life.

## 1. Introduction

The determination and management of the shelf-life of fruits and vegetables are affected by multiple factors that are usually hard to track, as most undergo changes that go beyond spoilage or contamination. Many of these products are sold in bulk and without any protective packaging, making them even more susceptible to loss and waste. The final quality of these perishable products depends not only on their pre- and post-harvest handling but also on the intrinsic biochemical characteristics that directly affect their ripening process and the organoleptic changes that ensue [1,2].

The traceability of these complex interactions between fruits and vegetables and their surrounding environment can be highly improved with the implementation of Artificial Intelligence (AI) systems [3], as they are particularly suited to intersect data from numerous sources and find relevant connections between them. There are indicators that the resulting data from these systems, commonly referred to as “smart data”, could be crucial for an integrated approach that will eventually lead to improvements in quality and waste prevention [4,5,6].

Computer vision (CV), which automates visual assessments, has been explored to enhance quality inspection tasks since the beginning of the century [7] but has only seen widespread adoption in recent years [3]. Advancements in CV systems for fruit quality assessments have shown promising results in variety identification and grading [8], defect detection [9], and even shelf-life estimations [5,10]. Most of them rely on convolutional neural network (CNN) models, especially suited for the recognition of patterns in visual data, that have experienced substantial growth in recent years. Inspired by the mammalian visual cortex, with different cells arranged in a layered architecture, these models can construct image segments at different levels of abstraction. This makes them highly effective in recognizing spatially dependent data, such as images and videos [11,12].

The avocado market has greatly benefited from their increasing association with a healthy lifestyle [13] and trending social media coverage [14]. It is anticipated that avocado production will continue to be the fastest-growing among major tropical fruits, with projections estimating an increase to 12 million tons by 2032—more than triple the output of 2010. This growth is driven by strong market demand and the fruit’s high export value, leading to considerable investments in both established and emerging production regions. Global avocado exports have reached quantities comparable to mangoes and are expected to surpass pineapple exports between 2028 and 2030, positioning avocados as the second most traded tropical fruit, only behind bananas. With its high average unit prices, avocados are projected to become one of the most valuable fruit commodities [15].

Currently, ‘Hass’ stands as the most sought-after avocado cultivar by a significant margin, with numerous new selections of ‘Hass’-like cultivars forming the foundation of avocado exports to the United States and the European Union (EU) [16]. In the EU, Portugal and Spain contribute to approximately 90% of avocado production [17], which, however, fulfills less than 20% of European consumption [18]. These limitations, common to most tropical and subtropical fruits, result in avocados passing through lengthy and time-consuming distribution channels, which complicates the management of their post-harvest handling. [19].

As a climacteric fruit, avocados do not ripen until after they are harvested, remaining mature yet unripe until picked [20,21]. The pre-harvest maturity significantly influences the post-harvest behavior of avocados and is commonly evaluated through dry matter analysis, which has been recently automated with portable near-infrared spectroscopy (NIRS) devices [22]. After harvest, pigment changes serve as a crucial indicator of ripeness for ‘Hass’ and other avocado cultivars, as the fruits transit from a light green when unripe to a purplish black when completely ripe [23]. These changes only partly stem from a minor decline in chlorophyll concentration in the fruit’s skin, which stabilizes early in the ripening process. More significantly, they are driven by external factors enhancing anthocyanin biosynthesis, notably cyanidin 3-O-glucoside. The synthesis of these pigments, crucial for developing purple/black pigmentation, is influenced both by growing conditions and the post-harvest ripening environment. Avocados ripened at higher temperatures tend to develop this pigmentation earlier and more intensely, whereas those ripened at lower temperatures may soften with minimal skin darkening [23].

Despite the widespread adoption of NIRS devices for assessing fruit maturity at harvest, there is a lack of non-destructive methods for accurately assessing the ripeness of avocados during post-harvest periods. Innovative synthetic sensors, designed to emulate the fruit’s morphology, have been developed, and are capable of documenting both environmental and physical stresses that avocados endure during post-harvest handling. It is envisaged that as these undergo the same treatment as the fruits themselves, they can provide real-time insights into their ripening state [24]. Research into hyperspectral imaging and other technologies as non-invasive means to determine flesh firmness and dry matter content has been conducted, providing potential insights into the avocado’s internal state. However, the reliability of these technologies remains incomplete, and their deployment is associated with significant costs [25,26].

Given the notable visual transformations that ‘Hass’ avocado fruits undergo through the ripening process, there is growing interest in employing Computer Vision (CV) predictive models. Certain methods focus on training models to predict the number of days until the fruits reach the end of their shelf-life [27], while other strategies involve categorizing the ripening process into distinct stages, such as *unripe*, *ripe*, and *overripe* [25,28,29], or even to predict the firmness of the fruit’s flesh using traditional machine learning algorithms [30]. The results suggest that the application of image processing and convolutional neural network (CNN)-based models could provide a cost-effective, non-invasive, high-accuracy methodology for Ripening Index classification, but it would require large amounts of samples, from different sources and storage conditions, to be able to create predictive models of these complex interactions [31].

A significant advantage of this approach is the CNN-based models’ ability to leverage previously acquired knowledge for easier adaptation to new datasets. This strategy, termed transfer learning [32], not only simplifies tailoring established models for estimating ripening and shelf-life but also promises models that progressively improve and broaden their application scope.

The significant heterogeneity in the avocado ripening process, which is yet not fully understood, poses a major challenge to developing ripening estimation technology [33]. Holistic approaches capable of tracing and correlating all pre-harvest and post-harvest factors affecting the fruit’s ripening behavior, employing innovative, non-destructive methods for on-chain real-time analysis, could lead to more consistent quality tracking, thereby reducing retail loss and household waste [34].

This study aims to examine the potential of these technologies in enhancing the tracking of avocado quality throughout their ripening process. The goal is to not only provide real-time estimates of their shelf-life in various storage environments but also to expand upon existing ripening assessments, identifying additional stages beyond the current state of the art [25,28,29], and offering a more detailed representation of the continuous biochemical transformations that influence their quality and organoleptic properties. The creation of a labeled image database that pairs avocado photos with quality information will also enable further research and innovation, potentially transforming the post-harvest management of this fruit, with a positive impact along the whole supply chain.

## 2. Materials and Methods

A total of 486 avocados (*Persea Americana* Mill. cv Hass) were obtained directly from a supplier based in Tavira, Portugal. All were sourced from a group of local producers and harvested on the same day in March 2022. The supplier reported that the fruits were harvested from multiple orchards, from a total of 650 ha, spanning from the east to the west regions of Algarve, Portugal. Their operations were certified with GlobalG.A.P, GRASP, and Tesco Nurture. The fruits were transported in a refrigerated truck at 5 °C and delivered to the Research Centre for Biotechnology and Fine Chemistry (CBQF) in Porto, Portugal, within the third day of post-harvest.

Upon reception, the avocados were thoroughly washed and scrubbed using a water-based solution containing 190 mg·L^−1^ of active chlorine from sodium hypochlorite and then rinsed according to the recommended guidelines [35]. Eight avocados were randomly selected for the initial dry matter assessment (destructive methodology), and the remaining 478 fruits were labeled, weighted, and sorted between ten boxes so that each box had a similar number of samples with evenly distributed weights.

### 2.1. Fruit Maturity at Harvest

An initial dry matter assessment was performed on 8 randomly selected avocados, according to the official method of analysis for fruit samples, published by AOAC International [36].

Three flesh samples were extracted from each fruit, placed in pre-weighed Petri dishes, and weighed to ascertain the total mass of each flesh sample. These samples were labeled, and their raw flesh masses were recorded.

The Petri dishes containing the flesh samples were then placed in a dehydrating oven and exposed to circulating hot air at 103 °C for 24 h. Post-dehydration, the samples were weighed once more, the weight of the empty Petri dishes was subtracted to determine the mass of the dehydrated flesh, and the dry matter content was calculated using the formula in Equation (1).
(1)$$DM\;\left(\%\right)=\frac{raw\;flesh\;mass\;(g)}{dehidrated\;flesh\;mass\;(g)}\times 100\%$$

The dry matter contents from the flesh samples of each fruit were averaged to obtain weighted values. The results from the studied fruits were compared with standard values utilized in the food industry for determining the appropriate harvesting maturity stage.

### 2.2. Storage

The samples were distributed into three storage groups. Two groups were subjected to a controlled environment, with stable temperature and relative humidity (RH), while a third was left at room temperature conditions to evaluate how fluctuations in the storage profile would affect the visual accuracy of the predictions. The three groups are detailed as follows:T10—192 samples stored at a controlled temperature of 10 °C and 85% RH;T20—143 samples stored at a controlled temperature of 20 °C and 85% RH;Tamb—143 samples stored at room temperature and RH.

The group stored at room temperature was left in a laboratory with natural daily oscillations in temperature that ranged between 15.8 °C and 21.7 °C, with an average of 18.7 °C and a standard deviation of 1.2 °C. Due to technical difficulties, it was not possible to track the RH of this environment.

These environments were designed to stimulate fruit ripening, rather than to replicate retail conditions, where fruits are typically stored at temperatures below 5 °C to decelerate the ripening process during transport, and subsequently warmed to 20 °C to hasten ripening [37]. The idea was to introduce sufficient variability in ripening characteristics into the models, allowing them to account for how they manifest under different storage conditions.

### 2.3. Image Collection

The samples were photographed daily in a HAVOX-HPB-40D Photo Studio Lightbox (42 × 42 × 42 cm) (Avolux SAS—Istanbul, Türkiye), with a matte white backdrop, illuminated by two LED ramps with a luminous flux of 12,000 lm ± 200 lm and a color temperature of 5500 K, paired with a light diffuser cloth. The distance between the fruits and the camera was 30 cm. The photos were obtained using a Canon EOS 60D DSLR camera, equipped with the Canon EF-S 18–55 mm f/3.5–5.6 IS II lens, mounted on top of the studio box. The following camera settings were used: ISO: 100; Aperture: f/8.0; Shutter Speed: 1/20 s. The photographs were collected using the EOS Utility 2 Software (Canon Inc.—Tokyo, Japan) and labeled according to the sample number, front or back position, storage group, and date.

### 2.4. Initial Color Analysis

As an additional assessment of the visual state of the avocados upon harvest, color analysis was performed using the image processing and Computer Vision Toolbox in MATLAB R2023b (MathWorks Inc.—Natick, MA, USA). All 956 photographs taken on the first day of the experiment were segmented to remove the background, and their chromaticity was extracted from the RGB values by applying calibration corrections using a Konica Minolta CR-400 colorimeter and a calibration kit (Konica Minolta Inc.—Tokyo, Japan) [38]. For each segmented picture, a k-means clustering technique was applied [39] that averaged the CIELAB values from 5 clusters, generating a 5-color palette. From the 956 generated palettes, 6 more predominant L* (lightness), a* (red-green component), and b* (yellow-blue component) values were registered, along with their corresponding average RGB values. These were then combined in a six-color palette, designed from left to right by order of predominance.

### 2.5. Data Labeling Methodology

Inspired by ripening classification methodologies for shelf-life estimation of ‘Hass’ avocados [25,28,29], a Ripening Index was designed that classified each sample between 1 and 5, corresponding to the following stages of the ripening process:Unripe;Breaking;Ripe (First Phase);Ripe (Second Phase);Overripe.

These were assessed by a team of two trained researchers and were based on a set of common visual and texture traits for each stage, which are detailed below.

Figure 1 shows five examples of the photographed samples, depicting the visual changes over the ripening process of Hass avocado fruits. These were chosen to help characterize the five stages of the Ripening Index.

The first stage (a) is characterized by a yellowish-green color and a very firm texture. The fruits might show signs of sun damage or other marks associated with their pre-harvest conditions. In the second stage (b), signs of ripening start to manifest in a darkening pigmentation, which should now be of a greyish olive green with hues of brown. It has a firm texture, though it should give slightly when pressed. In stage 3 (c), as the avocado fruit becomes ripe, shades of purple start to appear, scattered along the skin. Its texture is now less firm to the touch, signs of an easily sliced flesh, which should yet resist mashing. The fourth stage (d) is considered the last one of the fruit’s shelf-life, where its ripeness is at the maximum value but with no relevant signs of senescence or degrading quality. The fruit’s skin should now be of a homogenous purple shade, and its flesh easily displaced by a slight touch. The stem should appear dry and of a light brown color. In the fifth and last stage (e), the fruits show clear signs of senescence, with the appearance of mold spots throughout the skin and around the stem. In terms of texture, a separation between the exocarp and the mesocarp can be observed.

A 10-stage Ripening Index was also tested, where each of the original stages was divided in half, as depicted in Figure 2. As shown in the picture, the end of shelf-life corresponds to the 9th stage in this 10-stage index, which is considered the point where the characteristics of the fruit are no longer suitable for commercialization. The 10th and last stage is the one where evident signs of contamination would make it unsuitable for consumption.

For simplicity in visualization, each original stage is represented as being of equal size, although this was not necessarily the case in experimental observations. The stages are derived from noticeable visual and textural changes that occur during the ripening process of Hass avocados, rather than from precise divisions of its time frame.

### 2.6. Database

When photographing each sample, two researchers—previously trained in evaluating the Ripening Index of avocado fruits—assigned an index classification to each sample using both the 5-stage and 10-stage systems. This process contributed to the creation of a labeled image database. Each entry in the database consisted of an image paired with details about the sample, including the “sample number”, “storage group”, “side of the fruit”, “Ripening Index”, and a date stamp.

After the image collection was over, an additional piece of information, designated as *Days Left*, was added to each labeled image. This metric consisted of the number of days passed from the moment each sample was photographed until it would reach either a stage 5 classification of the 5-stage Ripening Index or stage 9 of the 10-stage index, considered as the endpoints of the avocado’s shelf-life. These data would later be used to estimate the expected shelf-life for any given classification within a particular storage group.

The database was divided into four distinct datasets. The first, labeled as “general”, contained images from all storage groups. This dataset was designed to assess the models’ capability to generalize their ripening stage predictions across various storage conditions, without being influenced by the specific effects of each storage environment on the ripening process [23,37]. The remaining three datasets were created for each storage group. Subsequently, each of the four datasets was duplicated, with one set labeled according to the 5-stage index and the other according to the 10-stage index, resulting in a total of eight final datasets.

Each dataset was divided into three subsets. The first subset, the training set, constituted 70% of the dataset and was used for successive iterations of the training process across the models. The validation set, forming 15% of the dataset, was used for periodic assessments, enabling consistent evaluation of the model’s generalization capabilities. This process facilitated the fine-tuning of training hyperparameters to enhance accuracy and minimize loss, ensuring that the model did not overfit the training data. The remaining 15% of the dataset constituted the test set, reserved for the ultimate evaluation of each model’s capability to predict and generalize based on the accuracy metric.

To prevent data leakage and ensure the models’ ability to generalize, each sample within a given dataset was uniquely assigned to either the training, validation, or test set, avoiding any overlap within those subsets [40]. Efforts were made to maintain a similar distribution of classification labels within each set to minimize bias. This was challenging due to the uneven occurrence of classifications throughout the experiment. Strategies to address this included employing subtle geometrical data augmentation techniques and conducting test runs with datasets adjusted for balance. In these adjusted datasets, samples from underrepresented classifications were randomly duplicated to ensure an equitable distribution of classifications within the training set, a technique known as random oversampling [41]. Considering the success of this method in the conducted test runs, it was implemented across all training sets but was not extended to the validation and test sets, which retained their original distribution without any duplication.

The final database comprising all acquired photographs and their respective classification labels was published on Mendeley Data, and both the pictures and respective labeling data can be accessed with the doi reference number 10.17632/3xd9n945v8.1.

### 2.7. Predictive Model Design and Architecture

Two well-established pre-trained convolutional neural network models were used as a base for designing the predictive models. The first, AlexNet, comprises five convolutional layers and three fully connected layers, making it an 8-layer deep model [42]. The second model, ResNet-18, is more advanced with a deeper architecture that includes 17 convolutional layers and one fully connected layer. ResNet-18 is notable for its use of skip connections, which enhance its efficiency in balancing depth with resource utilization [43]. These models were chosen for their proven effectiveness in image recognition [44], a critical factor given the limited computational resources available for training. Additionally, the choice facilitated an evaluation of how performance varies with the model’s complexity and depth. AlexNet, featuring an older and simpler architecture, contrasts with ResNet-18, which embodies a more contemporary design with significantly greater depth. Both the pre-trained AlexNet and ResNet-18 models have been employed in fruit quality assessments, yielding promising outcomes. Both stood out as the two most effective in recent investigations into the field [45,46].

The models were developed using the Deep Learning Toolbox in MATLAB R2023b (MathWorks Inc.—Natick, MA, USA), which features modular editing tools, enabling the adaptation of the pre-trained model’s base architecture to meet the needs of new datasets.

Modifications were applied to the final fully connected layer of each model to align its output with the number of classes in the new predictive model—ten classes for the 10-stage index and five for the 5-stage index. Similarly, the output layer, referred to as the classification layer, was updated to conform to these specifications.

### 2.8. Training, Testing, and Validation of the Predictive Model

The models were trained using a single CPU, an AMD Ryzen™ 7 5800H with 8 cores and a base clock of 3.2 GHz, with an integrated GPU AMD Radeon™ Graphics with 8 cores running at 2.0 GHz. Test runs were conducted to fine-tune the hyperparameters of each model. Following this optimization process, models were trained using datasets balanced for equitable distribution of classes, including over 30 epochs with a mini-batch size of 128 images. Geometric data augmentation techniques were applied, including random reflection, rotation (between −10° and 10°), rescaling (from 0.95 to 1.05), and horizontal and vertical translations (from −10 to 10 pixels). These techniques were selected to enhance model generalization [47]. However, considering the images in the database were already similar in scale and positioning, the adjustments applied were deliberately subtle. According to MathWorks Inc. Help Center documentation, “one randomly augmented version of each image is used during each epoch of training, where an epoch is a full pass of the training algorithm over the entire training data set. Therefore, each epoch uses a slightly different data set, but the actual number of training images in each epoch does not change” [48].

Models based on ResNet-18 were trained with an initial learning rate of 0.01, adjusted down by a factor of 0.1 every 10 epochs. This strategy aimed for rapid early convergence while minimizing the risk of overfitting in later stages. In contrast, the simpler architecture and absence of normalization layers in AlexNet-based models necessitated a lower initial learning rate of 0.001 to prevent overshooting the optimal loss [49]. A similar adjustment schedule of 0.1 reduction every 10 epochs was also applied. Validation checks were conducted every 10 iterations, with a validation patience setting of 150 iterations to prevent prolonged training without improvement. The final model was set to be the one that presented the best validation accuracy/loss. Figure 3 provides a visual representation of these divisions and the overall process.

## 3. Results

### 3.1. Initial Observations

Table 1 shows the initial dry matter content of the analyzed samples to help characterize the initial physiological characteristics of the fruits at their stages of maturity.

As an additional characterization of the fruits upon arrival, Figure 4 shows the six most predominant colors resulting from the extraction of the CIELAB color values from the 956 first-day pictures, corrected using the calibration technique. Such a color palette was found to better describe the chromaticity of the initial samples, as it shows the most predominant hues without losing too much information from averaging them out in a single output.

### 3.2. Database Descriptive Analysis

Table 2 provides a descriptive analysis of the database, specifically the number of photographs corresponding to each stage of the Ripening Index. Efforts were made to distribute these images across the training, validation, and test sets according to the 70%/15%/15% split specified in the methodology. Python script was used to ensure a randomized allocation of samples while preventing images from the same sample from being placed in different groups. This approach aimed for an optimal distribution, and despite minor deviations, the distribution closely aligns with the intended proportions.

### 3.3. Model Performance

Table 3 shows the overall average accuracy scores across all datasets for each of the Ripening Index classification systems. Both models achieved very similar performance, with a slight advantage for the models based on ResNet-18. Given these similarities, and to avoid redundancy, the detailed performance results shown further in this study will focus on the best-performing model for each dataset.

To allow for an overall perspective on the general performance of the trained models, Figure 5 shows a representation of the eight testing sets, illustrating that a significant number of incorrect predictions were unique to each model, not shared between them.

To establish a correlation between each Ripening Index and its inherent predictive shelf-life, regression models were studied for each storage group, considering the attributed classifications of each photographed sample, and the registered days it took from that point until reaching the stage corresponding to the end of its shelf-life. This new classification was called *Days Left* and was included in the database. After verifying an apparent linearity between the evolution of these two variables, and forcing the shelf-life estimations to be zero for the final stages of the Ripening Index, Equation (2) was used for the linear regression modeling for the 5-stage Ripening Index.
(2)$$Days\;Left=\alpha \times \left(Ripening\;Index-5\right)$$

Table 4 shows the coefficients of the linear regression models based on Equation (2), using the Ordinary Least Squares method, with a confidence interval of 95% for the coefficient *α*.

For the 10-stage Ripening Index, the end of shelf-life was defined by the 9th stage, as already depicted in Figure 2. Following the same rationale, the linear regression model is derived from Equation (3).
(3)$$Days\;Left=\alpha \times \left(Ripening\;Index-9\right)$$

Table 5 shows the coefficients of the linear regression models based on Equation (3), using the Ordinary Least Squares method, with a confidence interval of 95% for coefficient *α*.

Figure 6 provides a visual representation of the shelf-life estimations’ progression as predicted by the regression models. Each figure includes a side-by-side comparison of the 5-stage and 10-stage model estimates, aligned on opposite horizontal axes for a direct comparison of their timelines. Additionally, the plots feature the average shelf-life values observed for each Ripening Index, along with their respective standard deviations.

Figure 7 and Figure 8 present the accuracy of the top-performing model across different subsets, delineated by stage, for a detailed analysis of its efficacy. Given that each fruit sample was photographed from two opposing sides at each time point, there were instances where the model accurately identified the ripening stage of one image but not its counterpart. This discrepancy often arose from visual anomalies such as spots or marks on one side of the fruit, which could confuse the model’s predictions. To address these variations, Figure 7 and Figure 8 also detail the model’s performance on a per-sample basis, specifically highlighting cases where it successfully predicted the ripening stage of at least one side of a fruit sample.

Furthermore, the confusion matrices in Figure 9 and Figure 10 offer a detailed examination of each model’s performance by visualizing not only the correct predictions but also the incorrect ones. This is achieved by comparing the actual classifications with those predicted by the models.

To assess the performance of the shelf-life estimates, and to compare the ones carried out by the trained panel while attributing ripening classifications with the ones derived from the model’s predictions, the shelf-life estimation loss was studied (Equation (4)).
(4)$$Loss=\left|Estimated\;Shelf\;Life-Shelf\;Life\right|$$

Following the same rationale of Figure 7 and Figure 8, Table 6 shows the average shelf-life estimation loss for the attributed classifications and the ones predicted by the models. As two pictures were taken at each time for every sample, the average shelf-life estimation loss per sample considers only the best-performing prediction of both sides of the same sample to account for the model’s potential when more than one side is assessed per sample.

Figure 11, Figure 12 and Figure 13 feature 2D Kernel Density Estimation plots that represent the likelihood of the difference in days between the estimated and the observed shelf-life of the avocado samples, with a bandwidth setting of one day. By increasing the kernel bandwidth, these plots facilitate an easier examination of the error distribution and enable direct comparisons between attributed classifications (red) and model predictions (blue), and also between the 5-stage and 10-stage models [50]. Additionally, the average loss for each Ripening Index is depicted above the corresponding stage.

## 4. Discussion

The winter of 2021/2022 was the fourth warmest in Portugal since 1931, marking it with the highest average maximum temperature observed in the last 90 years. Additionally, it was recorded as the fifth driest winter during the same timeframe, receiving only 33% of the typical rainfall for that season [51]. The severe to extreme drought conditions that affected the Algarve region during that season were identified by the avocado fruit supplier as a significant factor accelerating the maturation of the fruits, reporting that the avocados were already at or beyond their optimal marketable levels of dry matter content at the outset of the experiment.

The analyzed samples demonstrated an initial dry matter content between 31% and 39%, surpassing the typically recommended harvest range of 25% to 30% [52]. This reinforces the fact that the avocados were harvested at a stage of maturity beyond the norm for suppliers, which is likely a consequence of the aforementioned atypical weather conditions. The observed high dry matter content indicates a potential for accelerated ripening, alongside a heightened susceptibility to physiological disorders and mechanical damage [53]. Notably, a considerable number of avocado samples exhibited wind rub damage, caused by the abrasion of the fruit’s skin against branches or leaves, alongside instances of sunburn damage. These factors introduced a level of visual variability in the damaged fruits that is atypical in market conditions, posing challenges to the generalization of the models.

As anticipated [23,37], the storage conditions significantly influenced the ripening process of the avocados. The fruits stored at 20 °C and at room temperature ripened approximately twice as fast as those kept at 10 °C. Moreover, the visual changes observed in the fruits varied with temperature, as fruits at higher temperatures developed their characteristic purplish pigmentation before any signs of senescence were noticed, which was not always the case for the fruits that had a longer ripening time frame. As a result of this, both Figure 7b and Figure 8b show a decrease in accuracy during the final stages of ripening for predictive models trained solely on samples stored at 10 °C.

Although the stages of the Ripening Index were defined considering variations in quality and organoleptic features of Hass avocados, the conversion of these stages into a shelf-life estimation demonstrated a clear linear progression, as evidenced by both the 5-stage and 10-stage models depicted in Figure 6. Nevertheless, the high degree of heterogeneity in the ripening of avocado fruits posed a constraint to the precise modeling of this progression, underscoring the necessity and potential for exploring smart tools capable of improving the traceability of quality in avocados.

The use of transfer learning [32], a key principle in this project, showed promising results, with both pre-trained models achieving comparable accuracy despite their architectural and complexity differences [42,43], with a slight advantage to ResNet-18. This observation suggests that using deeper and more complex models may offer limited scope for improvement in this context, which partially stems from not only the obvious limitations in ripening assessment based solely on visual information but also from the associated human error involved in hundreds of classifications every day; this could be reduced with improved systems but is hardly eliminated.

Nevertheless, a comparative analysis of both models’ performance, shown in Figure 5, revealed that the overlapping errors accounted for less than half of the total errors. This suggests that combining different models in an ensemble approach could potentially reduce the error rate. To achieve this, more in-depth studies of the strengths and weaknesses of each model would have to be carried out, so that these could be compensated by the consideration of different models or even different smart tools.

Examining the performance of the models across different ripening stages, as illustrated in Figure 7 and Figure 8, it is evident that the first stage consistently achieves the highest scores in almost every model, which could be attributed to two main factors. Firstly, the visual characteristics of the unripe stage appear to exhibit lower variance compared to other stages, making it more distinct and easier to identify. This distinctiveness is further enhanced as the unripe stage is less influenced by the storage environment due to the minimal accumulated storage time. Second, despite employing a random oversampling technique to balance the data and prevent the models from being biased towards the most frequently observed stage, this approach might still not adequately enhance the models’ ability to generalize across the less common stages of ripening [41].

The disparity in accuracy between per-picture and per-sample assessments, as shown in Figure 7 and Figure 8, underscores the impact of marks and spots on the model’s predictions. This distinction is crucial, especially considering the weather conditions affecting most samples, as previously discussed. An outlier removal system that averages the predictions across several images of the same fruit could help diminish this discrepancy, enhancing the overall prediction accuracy.

The confusion matrices depicted in Figure 9 and Figure 10 provide a more in-depth visualization of each model’s performance. The 5-stage models show higher accuracy on the extremes of the Ripening Index (with the discussed exception of the T10 dataset), whereas the performance of the 10-stage models fluctuates more throughout the ripening stages. This indicates that expanding the classification system to a 10-stage index, as anticipated, introduced greater complexity in both assigning and predicting the ripening stages. However, upon examining the distribution of the predictions, it becomes apparent that allowing for a 1-stage margin of error could be a decisive factor in favoring the 10-stage system. This is because over 93% of the 10-stage models’ predictions were within one stage of the attributed classifications, which demonstrates the benefit of the 10-stage system’s finer granularity, where each stage denotes a more specific phase of ripeness. Such precision ensures that a one-stage margin of error in the 10-stage system still offers a relevant predictive accuracy and practical utility—a level of usefulness not paralleled by the broader categories of the 5-stage system.

Developing a Ripening Index was a key strategy to translate the continuity of the ripening process in a sequential numerical index, a requirement of these smart systems. Determining the optimal number of stages for this index posed a significant challenge. While a 10-stage index appeared to offer precise ripening stage predictions for avocado samples, when given an error margin that could not be applied to the 5-stage index, analysis of the loss function for shelf-life estimations revealed similar average losses between the 10-stage and 5-stage models, as shown in Figure 11, Figure 12 and Figure 13. It is important to note that shelf-life estimation, although vital, is just one of the objectives of these smart tools. Defining a ripening stage provides immediate insight into the fruit’s condition at the time of the prediction, whereas shelf-life estimation only predicts the duration before spoilage signs emerge.

The experiment also highlighted the inherent unpredictability in the ripening process of avocado fruits, which posed challenges in generating accurate shelf-life estimations based on ripening classifications, particularly for samples stored at lower temperatures. Analysis of the discrepancy between the estimated and actual values showed that the loss distribution for both human-attributed classifications and model predictions was similar. This similarity suggests that the models performed on par with the trained panel in predicting the shelf-life of avocados.

As all the avocados used in this study were sourced from the same region and harvested at the same time, there was a small degree of variation in what concerns the pre-harvest conditions that could impact their ripening process. Although this allowed for a better comparison of the post-harvest handling factors, as any behavioral differences between sample groups could be attributed to their storage environment, conducting further research is essential to develop models that can be generalized to scalable applications.

The decision to select only two pre-trained CNN models as the foundation for the predictive model was primarily driven by the limited computational resources available for training 16 different models. Exploring other configurations through feature extraction assessments could lead to the development of more robust models. Furthermore, creating ensembles that leverage synergies from recent advancements in deep learning could significantly enhance the accuracy of ripening predictions and shelf-life estimations.

Additionally, considering the exploration of two classification indexes, each with its respective benefits and drawbacks, employing both simultaneously for assessment consistency and outlier detection could enhance the efficacy of this methodology. In the long term, diversifying the approach by training other models to predict various metrics might further minimize estimation errors and contribute to the development of a more comprehensive smart tool.

## 5. Conclusions

The application of pre-trained CNN models in predicting quality and shelf-life was explored using ‘Hass’ avocados by monitoring their ripening process throughout the entire post-harvest period. This involved daily photography of the fruits, coupled with quality information assessments. Improvements in the application of smart-data tools in the food industry require large amounts of data that can be channeled into their development. The creation of a labeled photographic database laid the groundwork for this project and, more importantly, opened the door for further exploratory studies that can leverage the collected data to develop new and enhanced tools.

Significant variation in the progression of Ripening Index classifications was observed within each storage group. This variability may partly stem from human error in the visual assessment. However, it was primarily attributed to the well-documented heterogeneity in the ripening of avocado fruits. While this variability challenges the development of smart-data tools for accurate monitoring, it also highlights the value of using machine learning and other computational methods that could provide a more effective way to understand and manage the complexities of avocado ripening.

The use of open access and highly efficient pre-trained CNN models was significant, as it demonstrated the feasibility of replicating these methodologies without the need for extensive computing resources. Furthermore, the potential for model ensembles to enhance performance suggests substantial opportunities for further improvement.

The findings from the implemented methodology underscore the potential to increase the robustness of the models by training them to identify additional quality features. These features can complement the existing assessments, aiming for a more holistic, real-time monitoring of avocado ripening and quality throughout post-harvest handling.

The potential applications of smart-data tools, designed to predict the shelf-life and assess quality-related factors of perishable products, are vast and will benefit from further research, which will only enhance their utility and effectiveness. The outcomes of this project underscore this trajectory, highlighting the transformative impact these tools could have on the food industry. Integrating such tools with real-time data analytics and other machine learning algorithms could further refine predictions, with the potential to transform the food supply chain and significantly reduce its associated loss and waste.

## Figures and Tables

**Figure 1 foods-13-01150-f001:**

Examples of stage 1 (**a**), stage 2 (**b**), stage 3 (**c**), stage 4 (**d**), and stage 5 (**e**) samples, classified according to the 5-stage Ripening Index.

**Figure 2 foods-13-01150-f002:**
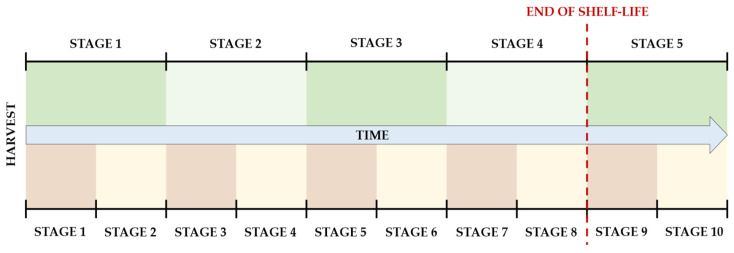
Scheme of the progression of the 5-stage Ripening Index (**top**) and the 10-stage Ripening Index (**bottom**).

**Figure 3 foods-13-01150-f003:**
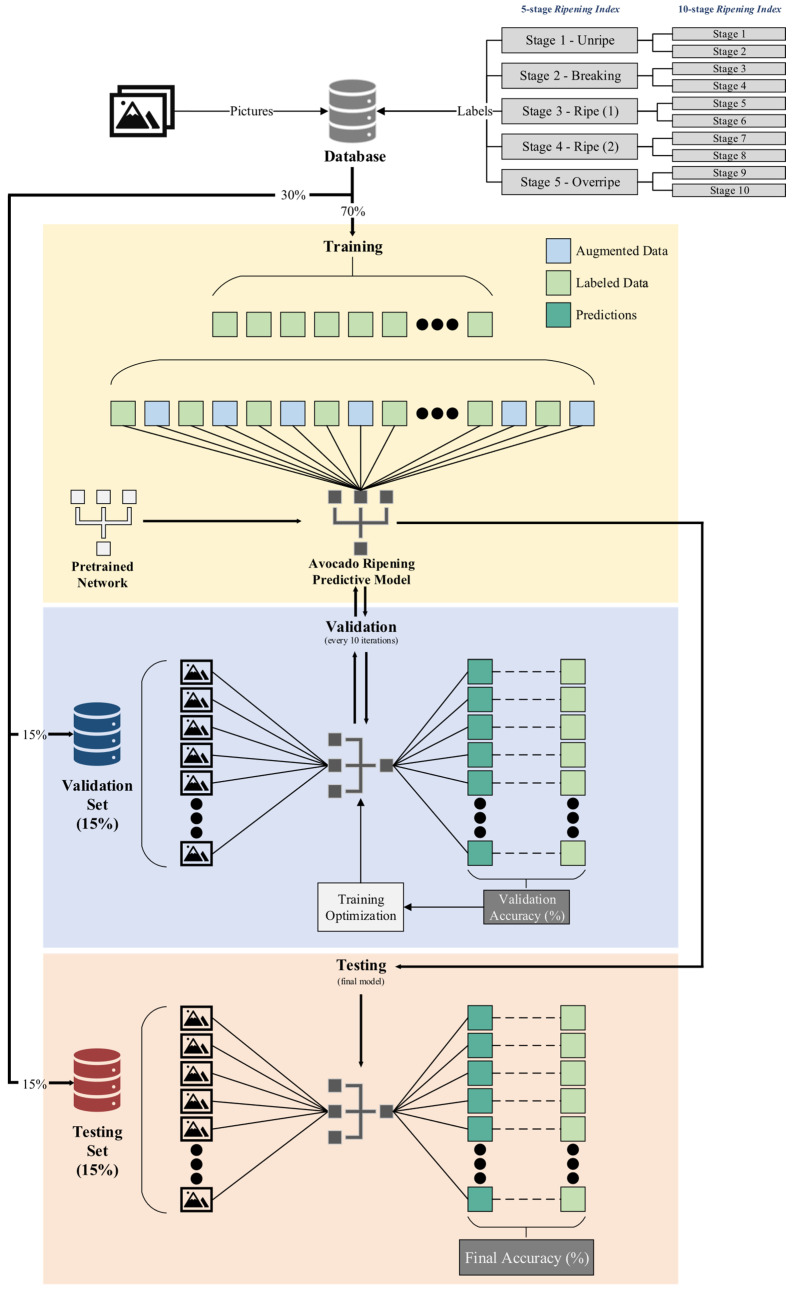
Scheme representing the building process of the predictive models, using the pre-trained convolutional neural network (CNN) models AlexNet and ResNet-18.

**Figure 4 foods-13-01150-f004:**
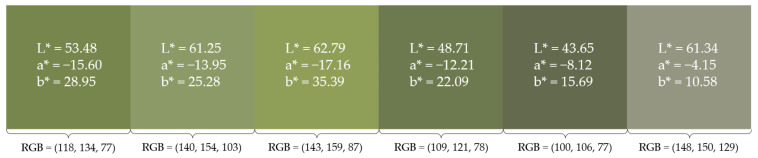
Color palette of the six most predominant colors of the avocado samples at the start of the experiment, corresponding RGB values, and corrected CIELAB values.

**Figure 5 foods-13-01150-f005:**
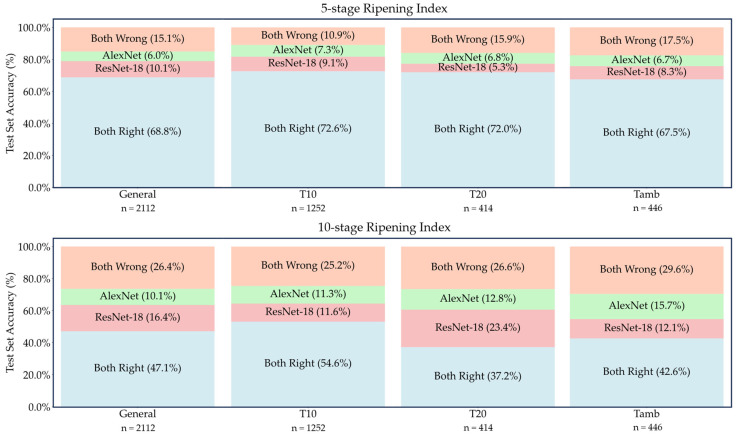
Model performance comparison for each test set accuracy with the 5-stage (**top**) and the 10-stage (**bottom**) classification systems.

**Figure 6 foods-13-01150-f006:**
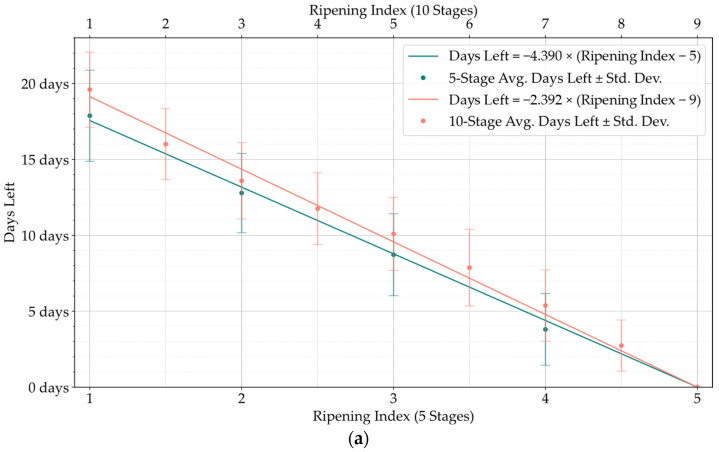
Linear regression of the ‘Days Left’ of shelf-life in each stage of the 10-stage Ripening Index (**top axis**) and the 5-stage Ripening Index (**bottom axis**), and scatter plot of the average shelf-life in each stage of the T10 (**a**), T20 (**b**), and Tamb (**c**) storage groups.

**Figure 7 foods-13-01150-f007:**
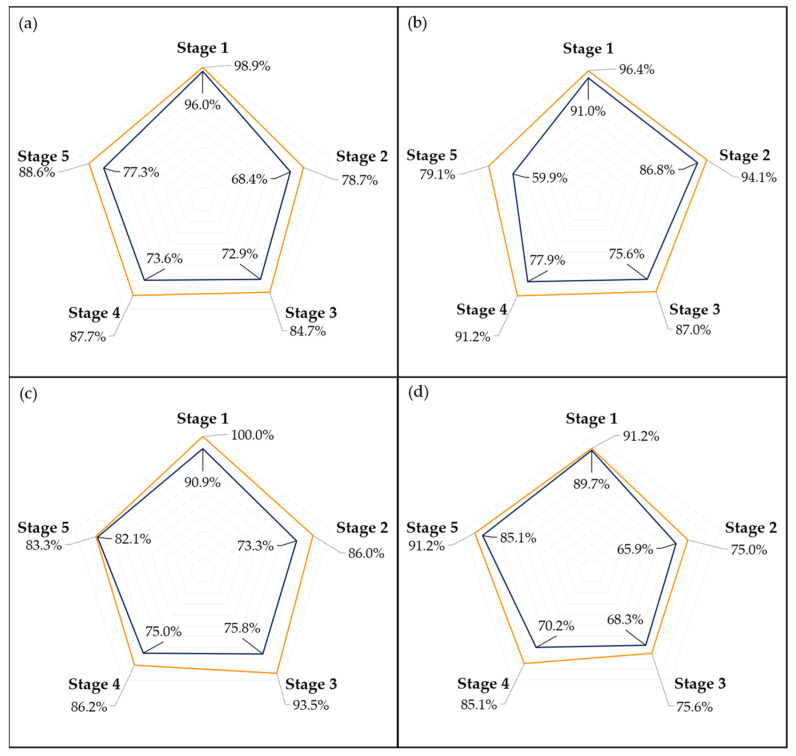
Accuracy score by picture (blue) and by sample (orange) for the best-performing network on the general (**a**), T10 (**b**), T20 (**c**), and Tamb (**d**) datasets, on a 5-stage Ripening Index.

**Figure 8 foods-13-01150-f008:**
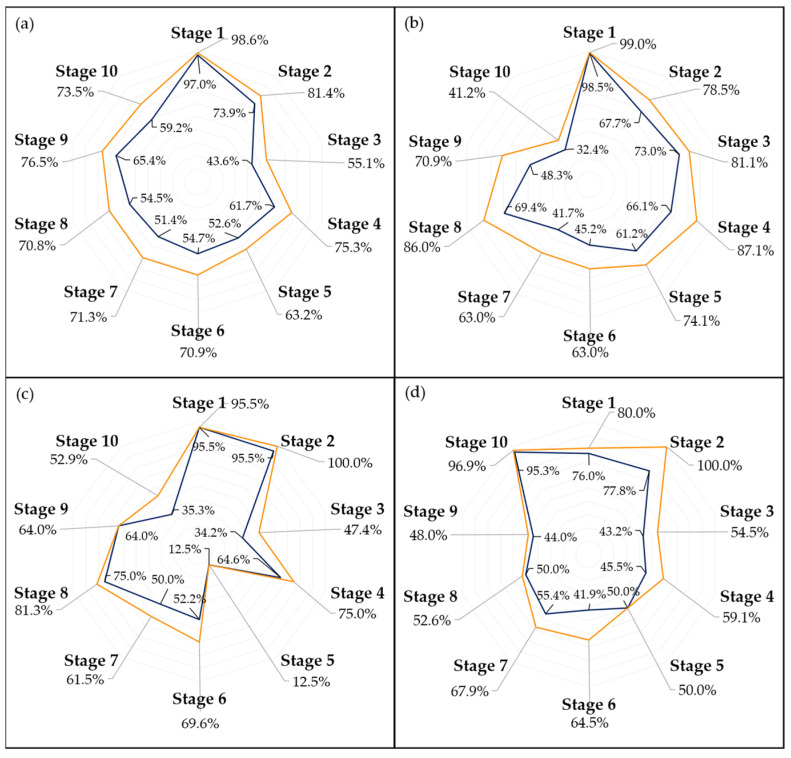
Accuracy score by picture (blue) and by sample (orange) for the best-performing network on the general (**a**), T10 (**b**), T20 (**c**), and Tamb (**d**) datasets, on a 10-stage Ripening Index.

**Figure 9 foods-13-01150-f009:**
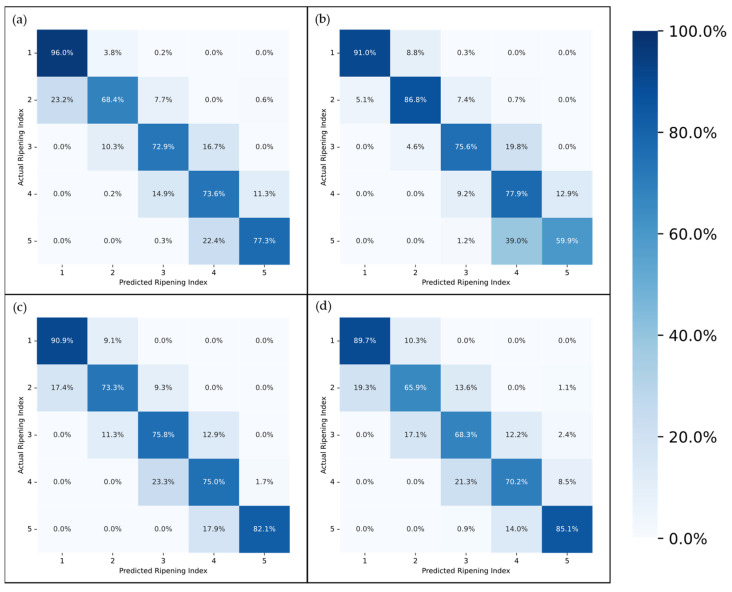
Confusion Matrix for the best-performing network on the general (**a**), T10 (**b**), T20 (**c**), and Tamb (**d**) datasets, on a 5-stage Ripening Index.

**Figure 10 foods-13-01150-f010:**
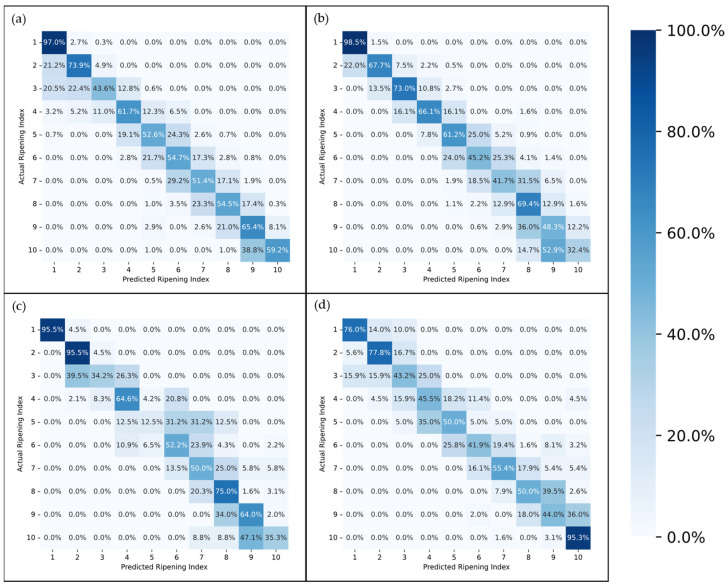
Confusion Matrix for the best-performing network on the general (**a**), T10 (**b**), T20 (**c**), and Tamb (**d**) datasets, on a 10-stage Ripening Index.

**Figure 11 foods-13-01150-f011:**
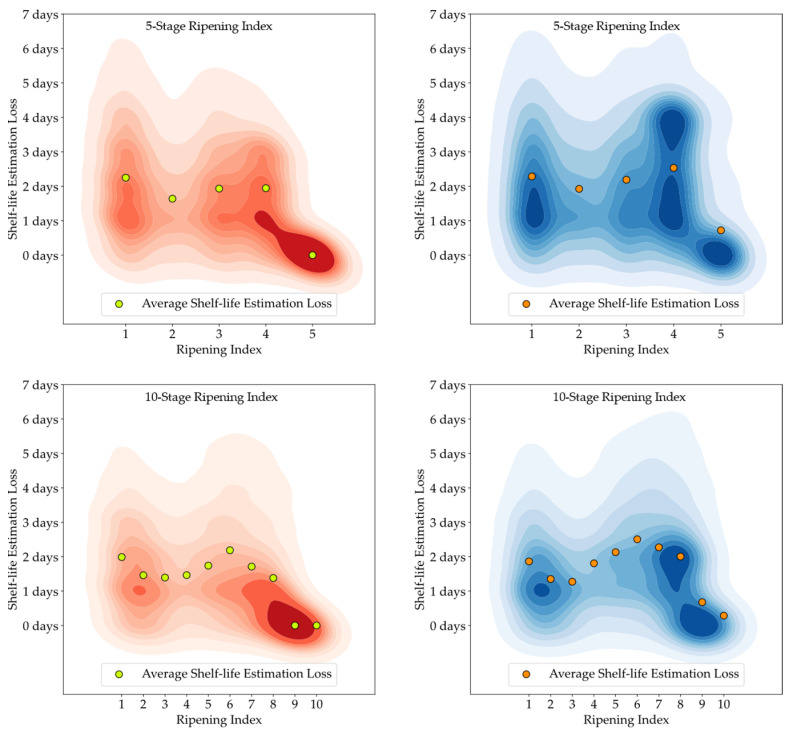
Kernel Density Estimation plots for the loss values of the estimated shelf-life, based on attributed classifications (red) and model predictions (blue), for the T10 storage group.

**Figure 12 foods-13-01150-f012:**
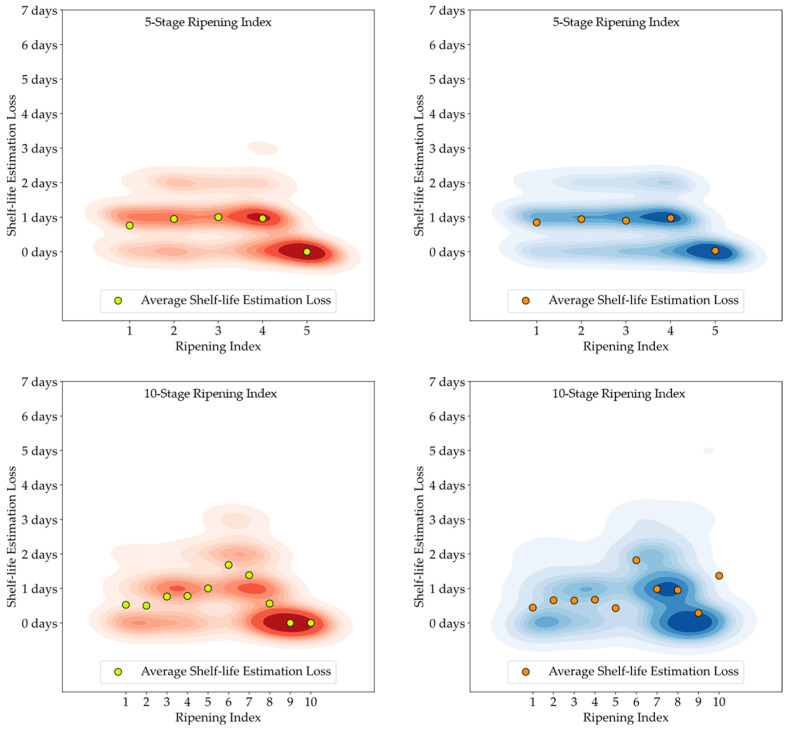
Kernel Density Estimation plots for the loss values of the estimated shelf-life, based on attributed classifications (red) and model predictions (blue), for the T20 storage group.

**Figure 13 foods-13-01150-f013:**
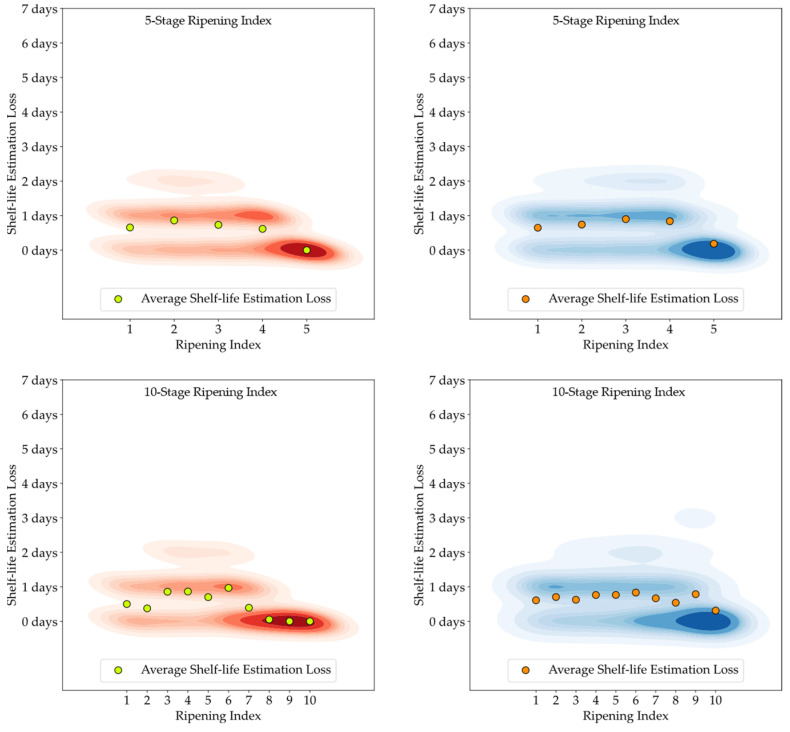
Kernel Density Estimation plots for the loss values of the estimated shelf-life, based on attributed classifications (red) and model predictions (blue), for the Tamb storage group.

**Table 1 foods-13-01150-t001:** Initial dry matter (D.M.) of the analyzed avocado samples.

Average D.M.	34%
Std. Deviation	2.8%
Maximum D.M.	39%
Minimum D.M.	31%

**Table 2 foods-13-01150-t002:** Descriptive analysis of the final database.

Ripening Index	Training Set	Validation Set	Test Set	Total
1	1402	298	308	2008
2	1100	228	236	1564
3	808	162	180	1150
4	778	154	152	1084
5	688	152	142	982
6	1246	256	274	1776
7	1042	216	198	1456
8	1290	288	260	1838
9	1508	272	392	2172
10	488	98	106	692
Total	10,350	2124	2248	14,722

**Table 3 foods-13-01150-t003:** Average accuracy scores for networks derived from two pre-trained models (AlexNet and Res-Net-18) across all storage groups.

Pretrained Network	Ripening Index	Margin of Error	Average Score ^1^
AlexNet	5 Stages	None	76.9%
		1 Stage	99.4%
	10 Stages	None	57.4%
		1 Stage	92.6%
		2 Stages	98.4%
ResNet-18	5 Stages	None	78.4%
		1 Stage	99.7%
	10 Stages	None	60.8%
		1 Stage	93.3%
		2 Stages	98.6%

^1^ Each trained model was considered with an equal weight to the average score, despite having different total occurrences.

**Table 4 foods-13-01150-t004:** Linear regression model coefficients for each storage group, based on Equation (2), applied to the 5-stage index classifications for shelf-life estimation.

Storage Group	Coefficient (*α*)	R-Squared
T10	−4.390 ± 0.021	0.953
T20	−2.116 ± 0.016	0.963
Tamb	−1.929 ± 0.015	0.959

**Table 5 foods-13-01150-t005:** Linear regression model coefficients for each storage group, based on Equation (3), applied to the 10-stage index classifications for shelf-life estimation.

Storage Group	Coefficient (*α*)	R-Squared
T10	−2.392 ± 0.011	0.964
T20	−1.156 ± 0.008	0.966
Tamb	−1.045 ± 0.008	0.965

**Table 6 foods-13-01150-t006:** Average shelf-life estimation loss for the attributed classifications and for predicted classifications, by picture and by sample.

Storage Group	Average Shelf-Life Estimation Loss ^2^		
	Attributed Classifications	Predicted Classifications (by Picture)	Predicted Classifications (by Sample)
T10 (5 stages)	1.70 days	1.94 days	1.82 days
T20 (5 stages)	0.73 days	0.75 days	0.75 days
Tamb (5 stages)	0.53 days	0.64 days	0.54 days
Overall (5 stages)	0.99 days	1.11 days	1.04 days
T10 (10 stages)	1.40 days	1.74 days	1.50 days
T20 (10 stages)	0.73 days	0.74 days	0.66 days
Tamb (10 stages)	0.46 days	0.59 days	0.49 days
Overall (10 stages)	0.86 days	1.02 days	0.88 days
Overall	0.92 days	1.07 days	0.96 days

^2^ Each storage group was considered with an equal weight to the overall average score, despite having different total occurrences.

## Data Availability

The original contributions presented in the study are included in the article, further inquiries can be directed to the corresponding author. The data presented in this study are openly available in Mendeley data at https://data.mendeley.com/datasets/3xd9n945v8/1 (accessed on 7 April 2024).

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
