# Peer review of "Shelf-Life Management and Ripening Assessment of ‘Hass’ Avocado (Persea americana) Using Deep Learning Approaches"

_foods, 2024, doi:10.3390/foods13081150_

Round 1
Reviewer 1 Report
Comments and Suggestions for Authors
This research categorized the ripening stages of avocados to form a database of labeled images and two convolutional neural network models, AlexNet and ResNet-18, were trained through transfer learning techniques to identify distinct ripening indicators, enabling the prediction of ripening stages and shelf-life estimations for new unseen data. From my point of view, the manuscript is impressively clear and I recommend it minor revisions.
1. Line115-117: The authors should have supplemented the relevant indicators at fruit harvest, such as chromaticity L*, a*, b*, and averaged RGB, to reflect the initial physiological characteristics of the fruit at that stage of maturity.
2. Line121: “ppm” is ambigous unit. It is recommended that the authors use “mg L-1” or “%” to express sodium hypochlorite concentration instead of “ppm”.
3. Line149-156: How many images were captured in each stage?
4. Line327: “Figure 5Figure 7” should be represented as “Figure 5-7”. And check for correct expression of cited images in the manuscript.
5. Combine Figure 5, Figure 6, and Figure 7, and then mark them as (a), (b), and (c), respectively.
6. Line377: Please check the correct image hyperlink. Delete “Error! Reference source not found”.
7. Line592, 604: “Persea Americana” should be written in italic.
8. On reference part, please confirm the abbreviation of the journal name, and please revise the reference format based on the author's guideline or journal publication regulation.
Comments on the Quality of English Language
Minor editing of English language required
Reviewer 2 Report
Comments and Suggestions for Authors
The article ‘ Shelf-life management and ripening assessment of ‘Hass’ avocado (Persea americana) through deep learning approaches ‘ presented for review is interesting. The article is an original scientific study. The topic is of great practical and economic importance. However, some issues need to be clarified or supplemented. The comments are included below.
- The title is worded correctly and accurately reflects the content.
- The abstract is clear and adequate.
- In the reviewer's opinion, it is worth mentioning more about non-invasive methods of assessing fruit quality in the introduction. This is particularly important in the context of the topic discussed by the authors.
- In my opinion, some of the information contained in the data labeling methodology section should be included in the discussion subsection. I propose to reword this subsection.
- In my opinion, assessing the degree of ripening solely on the basis of photos showing the external appearance of the fruit is discussion . This assessment does not reflect changes in the firmness and structure of the fruit. Please explain why the authors limited themselves to assessing only this one parameter. There are also other non-invasive methods that provide much more information. The assessment of the degree of maturity does not fully reflect the technological suitability of the raw material. For example currently, devices for rejecting soft and unripe fruit using infrared photography are available on an industrial scale.
- Line 377 – Error
- Correct conclusions, closely related to the research conducted.
Reviewer 3 Report
Comments and Suggestions for Authors
The manuscript explored the deep learning approaches to monitoring avocado shelf life and ripening. The research provides a promising method to estimate the fruit quality and serve a lot in fruit production. The manuscript can be minor revisions, please see the below comments.
Regarding the experiment design, this study only has one replication from March 2022. However, fruit ripening and shelf life are influenced by multiple factors, which will make fruit quality vary a lot even from the same orchard. Such as the rainfall, temperature, and tree condition, before the harvesting. Therefore, it is debatable if the authors build a database just depending on one replicate. Ethier more fruit from other replicates, or use more other fruit to certify the accuracy of the model is recommended.
This study also lacks using real fruit to verify the applicability of the model and needs to measure fruit quality indicators to compare with the predicted shelf life of the model.
Line 115-119, the authors should provide the horticultural condition of the fruit and need to introduce how the fruit was collected, such as from how many trees.
Line 132, please provide the temperature and RH of the ambient environment.
Line 139, what was the distance between the camera and the fruit?
Line 162-174, did the author compare with commercial standards to identify the fruit ripening index? Such as how exactly the firmness or soluble solid content.
Line 377, reference errors.
